# Morphogenetic Designs, and Disease Models in Central Nervous System Organoids

**DOI:** 10.3390/ijms25147750

**Published:** 2024-07-15

**Authors:** Minsung Bock, Sung Jun Hong, Songzi Zhang, Yerin Yu, Somin Lee, Haeeun Shin, Byung Hyune Choi, Inbo Han

**Affiliations:** 1Department of Neurosurgery, CHA Bundang Medical Center, CHA University, Seongnam-si 13496, Republic of Korea; minsungbock@gmail.com (M.B.); szzhang95@gmail.com (S.Z.); dhdtm33@naver.com (Y.Y.); dmddmdmd5@gmail.com (S.L.); tlsgodms223@naver.com (H.S.); 2Research Competency Milestones Program, School of Medicine, CHA University, Seongnam-si 13488, Republic of Korea; sjhong1021@gmail.com; 3Department of Medicine, School of Medicine, CHA University, Seongnam-si 13496, Republic of Korea; 4Department of Biomedical Science, Inha University College of Medicine, Incheon 22212, Republic of Korea; bryan@inha.ac.kr; 5Advanced Regenerative Medicine Research Center, CHA Future Medicine Research Institute, Seongnam-si 13488, Republic of Korea

**Keywords:** central nervous system, organoids, cerebral organoids, morphogenesis, disease models

## Abstract

Since the emergence of the first cerebral organoid (CO) in 2013, advancements have transformed central nervous system (CNS) research. Initial efforts focused on studying the morphogenesis of COs and creating reproducible models. Numerous methodologies have been proposed, enabling the design of the brain organoid to represent specific regions and spinal cord structures. CNS organoids now facilitate the study of a wide range of CNS diseases, from infections to tumors, which were previously difficult to investigate. We summarize the major advancements in CNS organoids, concerning morphogenetic designs and disease models. We examine the development of fabrication procedures and how these advancements have enabled the generation of region-specific brain organoids and spinal cord models. We highlight the application of these organoids in studying various CNS diseases, demonstrating the versatility and potential of organoid models in advancing our understanding of complex conditions. We discuss the current challenges in the field, including issues related to reproducibility, scalability, and the accurate recapitulation of the in vivo environment. We provide an outlook on prospective studies and future directions. This review aims to provide a comprehensive overview of the state-of-the-art CNS organoid research, highlighting key developments, current challenges, and prospects in the field.

## 1. Introduction

The central nervous system (CNS), consisting of the brain and the spinal cord, involves a complex interplay of cell types that interact in a precise and controlled manner to form a complex neuronal network. Defects that occur during CNS development can lead to critical conditions, such as microcephaly, lissencephaly, autism, and developmental epilepsy. The CNS is constantly exposed to numerous risk factors and is vulnerable to various diseases [1,2,3,4]. Therefore, understanding the developmental process, physiology, and pathogenesis of the CNS is essential to uncover the underlying mechanisms, develop medicines, and prevent diseases.

Traditional in vitro and in vivo cell experiments on the human CNS fail to replicate its complicated environment accurately [5,6]. Direct studies on disease origin and progression in the CNS in patients face clear limitations. Non-invasive techniques, although widely used, fall short in studying early-stage disease progression [7]. Additionally, ethical and technical challenges make it difficult to study the developmental process of the human CNS from the embryos.

Accurate human CNS models are needed to identify CNS-relevant genotypes and phenotypes, both normal and abnormal. CNS organoids, which contain progenitor cells and differentiated cell types, provide powerful tools for studying CNS development, function, and disease pathology.

Organoids, primarily derived from pluripotent stem cells (PSCs), are transforming the research of human organ development, physiology, pathology, and pharmacology [8,9,10]. Recent studies have shown that organoids can serve as disease models and can be applied to drug screening and direct therapeutic interventions [11,12,13,14,15,16,17,18]. For example, Sato et al. [15] demonstrated stable epithelial organoids capable of differentiating into various organs. Crespo et al. [13] conducted successful drug screening on organoid-resembling colorectal cancer derived from induced PSCs (iPSCs). Jee et al. [14] showed that the implantation of colon organoids could be used as regenerative medicine in mouse models with radiation proctitis. Similar studies have been conducted in the kidney [16,17] and liver [11,12].

Despite these advancements, CNS organoids were less studied due to their complexity. Initial efforts primarily involved self-organized neural tissues derived from embryonic stem cells (ESCs) for in vitro CNS synthesis [19]. While effective in describing the early-stage development of individual cells, these systems lacked insight into late developmental processes involving specific regional identities. In 2013, the Lancaster group developed a human pluripotent stem cell-derived three-dimensional organoid culture system, known as cerebral organoids (COs), which exhibited the features of human cortical development [20]. The developed COs were heterogeneous and difficult to reproduce. There is a need to define the variables, such as the type of cell within an organoid, and the cell origin and media, that affect organoid homogeneity and reproducibility [21,22,23]. Studies have implemented a bulk analysis as a proof to show the consistent generation of organoids [24,25]. Despite the need for the further development and standardization of COs, the breakthrough spurred further research and development in diverse brain organoids [26].

The protocol for differentiating spinal organoids remains less established than that of COs due to the complex morphogenesis. Lee et al. [27] produced human spinal cord-like organoid and screened antiepileptic drugs, while Gribaudo et al. [28] demonstrated a human trunk-like organoid, presenting both asymmetric axial growth and anteroposterior patterning. Consequently, CNS organoids have emerged as novel models for exploring CNS development and CNS-related diseases in vitro.

In this review, we describe the conventional methodologies and recent technologies for developing CNS organoids based on architectural regions and discuss the current limitations that need to be overcome. Specifically, we summarize their potential applications in disease modeling and prospective future directions for CNS organoids developments.

## 2. Morphogenetic Design of CNS Organoids

During CNS development, complex cellular and genetic signaling systems differentiate stem cells into specific structures. In the embryonic stage, epidermal cells migrate to form primitive streaks, leading to regional neural progenitors. Early signals direct cells to the forebrain, while later signals guide cells to the hindbrain and spinal cord [29]. Cells exhibit characteristics based on morphogens, with large collectives self-organizing into functional units [30]. CNS organoids help us to understand CNS structure development and function, providing insights into CNS diseases. Accordingly, many researchers are developing CNS organoids with specific properties. In the section, we explain how organoids represent each CNS region and introduce research trends in generation protocols.

### 2.1. Cortical Organoid

In 2013, Lancaster et al. [20,31] successfully created the first CNS organoid, using embryonic bodies to induce neuroectodermal tissues [32], which were cultured in 3D scaffolds and then in bioreactors to form COs(Figure 1). Early COs were heterogeneous, displaying markers for various brain regions, including the forebrain, hindbrain, hippocampus, and cortex. The cortical layer and radial glia (RG) were of particular interest due to their significance in differentiating human and rodent brain structures. The morphology and orientation of RG in COs accurately reproduced the human neurodevelopmental process [33]. The prolonged culture of COs presented diverse cells, including those from the cerebral cortex [34]. A single-cell RNA sequencing (scRNA-seq) analysis revealed that six-month-old organoids exhibited various neuron types found in the endogenous human cerebral cortex.

This breakthrough spurred extensive research on COs and the CNS organoids. However, early COs lacked reproducibility, specificity, and reliability. Lancaster et al. [35] improved these aspects using floating microfilaments as scaffolds, enhancing the surface-to-volume ratio of the embryo bodies, which led to more reliable and reproducible COs (Figure 2). Observations of cortical plate formation advanced the understanding of cortical growth. Brain organoids could be generated with cell components such as the human cerebral cortex [36], showing markers for dorsal forebrain progenitors and early neurons at various stages of development. Further scRNA-seq on 166,000 cells from 21 organoids demonstrated consistent cellular components and developmental trajectories, underscoring the importance of the environment for the reproducibility and viability of organoids.

Cortical organoids are valuable for unraveling human-specific brain development (Table 1). Pollen et al. [37] investigated brain evolution using organoids from humans, chimpanzees, and macaque, finding 261 genes that are differently expressed in humans. Key pathways, such as phosphoinositide 3-kinase (PI3K), protein kinase B (AKT), and the mammalian target of rapamycin (mTOR), were upregulated in human brain organoids. The sliced neocortical organoid system enabled studies of deeper organoid parts, preventing cell death during long-term culture and allowing neurogenesis resembling the embryonic human neocortex of the third trimester (Figure 2) [38]. The role of Wingless/Integrated (Wnt)/β-catenin signals in cortical neuron specification was clarified. Uzquiano et al. [39] established an atlas of cortical development using organoids and a single-cell-level analysis, demonstrating molecular trajectories and early-stage transcriptional diversity, validating organoids as a suitable source for studying cortical development.

### 2.2. Forebrain Organoid

The forebrain, comprising functional units such as the thalamus, hypothalamus, and limbic system, is crucial for understanding neuronal diseases.

Research on the regional specification of forebrain organoids (FOs) aims to elucidate the developmental process and potential applications (Table 2). Forebrain organoids can display ventral and dorsal regions within continuous neuroepithelia, with the patterning influenced by Wnt and bone morphogenetic protein (BMP) signaling [40]. Krefft et al. [41] developed a reproducible method to culture early dorsal telencephalic tissues using self-organizing iPSCs and Suppressor of Mothers Against Decapentaplegic (SMAD) signaling combined with Wnt inhibition.

Dorsal and ventral FOs have been fused to study differentiation [42]. Ventral organoids, created by combining Wnt inhibition and sonic hedgehog (Shh) enhancement, and dorsal organoids co-cultured together exhibited active neuronal migrations, particularly GABAergic interneurons. Cederquist et al. [43] discovered a positional identity within continuous neural tissue, with spatial organization resembling in vivo models when Shh hPSCs aggregates were positioned at a pole. An advanced protocol for generating forebrain organoids with visual systems, termed the “optic vesicle-containing brain (OVB) organoids” was proposed [44]. These organoids, treated with a special medium, allowed the study of sensory organ and brain interactions and early visual neuron development defects.

Efforts to generate diencephalon-type FOs, including hypothalamic organoids, involved protocols for mouse hypothalamic organoids adapted to hiPSCs [45]. Hypothalamic patterning used Shh activation and Wnt inhibition during neural ectoderm specification, further maturing in the medium with hypothalamic astrocytes and trophic factors to create “arcuate organoids (ARCOs)”. scRNA-seq revealed significant molecular overlap between ARCOs and the human hypothalamus [46,47]. Sarrafha et al. [48] used a similar protocol to generate hypothalamus organoids, focusing on dopaminergic neurons by integrating tyrosine hydroxylase (TH)-TdTomato reporters.

The protocols for thalamus-like organoids aimed to elucidate thalamic development [49]. These organoids were fused with cortical organoids to study the reciprocal projections and tracts between the thalamus and the cortex, demonstrating axon targeting and synaptogenesis. Thalamic and ventral thalamic organoids, generated by controlling Shh exposure, displayed GABAergic nuclei, characteristic of the thalamic reticular nucleus in vivo [50].

### 2.3. Midbrain Organoid

The midbrain is crucial for connecting vestibulocochlear and visual organs to the brain and contains key nuclei for visual reflexes and dopamine production, particularly in the ventral tegmental area and substantia nigra [51]. The establishment of midbrain organoids (MOs) is essential for understanding and treating dopamine-related diseases (Table 3).

Jo et al. [52] first generated a human midbrain-like organoid from hPSC, and Monzel et al. [53] constructed organoids from neuroepithelial stem cells. These MOs contained spatially organized groups of dopaminergic neurons, characterized for neuronal, astroglia, and oligodendrocyte differentiation, synaptic connections, and electrophysiological activity. The MOs demonstrated midbrain functionality in vivo by showing decreased neuronal activity after dopamine treatment, indicating their potential use in Parkinson’s disease models. Both teams’ MOs included neuromelanin-producing neurons.

Specific patterning with SMAD inhibitors and a Wnt gradient was key to maintaining the MOs’ homogeneity [54]. Testing with small molecules for dual SMAD inhibition led to reproducible and standardized organoids, with functionality and cellular compositions comparable with developing human midbrains. Nickels et al. [55] established a protocol using three patterning strategies related to different culture times and cell densities, achieving MOs with a high viability and minimal necrotic cores. Berger et al. [56] further improved this by using a millifluidic culture system, enhancing the oxygen and nutrient supply to reduce the necrotic region and improve dopaminergic neuron differentiation (Figure 2).

An automated one-step workflow for generating and analyzing MOs was suggested, using a 96-well system for seeding, imaging, and analysis [57]. This automated procedure resulted in homogenous MOs, regarding the size, cellular composition, and phenotype.

### 2.4. Cerebellar Organoid

The cerebellum is crucial for coordination and detailed movements [58]. Muguruma’s group [58,59,60,61] is at the forefront of cerebellar organoid research. They differentiated human embryonic stem cells (hESCs) into neuroectoderms and treated them with fibroblast growth factor 2 (FGF2) to inhibit forebrain markers. FGF2 stimulated FGF8 and Wnt1 expression, which is essential for early cerebellar development [61]. This initiated a feedback loop promoting engrailed *homeobox 2* (*En2*) and *paired box gene 2* (*Pax2*) expression, leading to hESC aggregates displaying early cerebellar characteristics, including Purkinje cell differentiation [62]. Further treatment with FGF19 and stromal-derived factor 1 (SDF1) resulted in self-organizing cerebellar organoids resembling early embryonic cerebellum morphogenesis, with Purkinje cells and other GABAergic neurons.

Alternative methods were suggested to understand the developmental processes. Single bioreactors use a controlled size and shape, with the agitation speed governing the early-phase aggregate size [63]. This method induced the differentiation of major cerebellar neurons, retaining organoids for 3 months. Another study used a human-only air–liquid interface culture, achieving cerebellar neuron differentiation without SDF1 and FGF19 [64].

Chen et al. [8] developed a protocol maintaining cerebellar organoid quality for over 200 days, including deep cerebellar nuclei and glial cells. A multi-electrode array analysis confirmed complex neuronal activity and Purkinje cells’ role in cerebellar circuits. Another study reproduced the cellular composition and spatial characteristics of the human cerebellum in vivo [65], employing altered basal media to ensure cerebellar identity. These cerebellar organoids showed functional connectivity between the inhibitory and excitatory neurons within the multilayered structures mirroring the human cerebellar cortex. These studies enhanced our understanding of the human cerebellum at a cellular level, offering insight into cerebellar development and connectivity with the brainstem (Table 4).

### 2.5. Spinal Cord Organoid

Spinal cord organoids (SCOs) are challenging to generate due to their complex 3D structure and diverse subneurons within the tissue, as well as their intricate growth factor interactions (Table 5) [66,67,68,69].

Proper growth signaling treatment in the neuromesoderm activates homeobox (HOX) expression in a colinear pattern [70], which is crucial for designing initial human neural tube models from ESC. SCOs with dorsal and ventral identity were generated using BMP4 and Shh signaling, but these were not fully representative of the dorso–ventral axis [71]. Andersen et al. [72] achieved a continuous dorso–ventral axis using the cross gradients of BMP-4 and Shh, leading to motor neurons connection with muscle spheroids.

The dynamic control of FGF signaling manipulated *HOX* gene expression, differentiating motor neuron subtypes along the rostro–caudal axis [73]. Another study reproduced human somatogenesis using the FGF and WNT gradient with retinoic acid [74]. Chip-based culture systems addressed organoid heterogeneity by starting with patterned 2D stem cells on a chip, promoting neural progenitor self-organization into tubular structures, increasing the differentiation efficiency and reproducibility [75]. Lee et al. [27] demonstrated an SCO with a tubular structure mimicking human neural tube morphology, containing glial cells and functional synaptic networks, which is useful for drug screening [76]. However, these models did not fully recapitulate human spinal cord co-morphogenesis.

Gribaudo et al. [28] developed a self-organizing platform mimicking human spinal cord and spine development, autonomously forming the AP axis, and resembling in vivo conditions during later differentiation stages. This model identified various regulatory factors but exhibited inconsistencies in neural tube organization and segment arrangement across batches. Sophisticated fabrication methods for SCOs need further development for homogeneity and standardization.

**Table 5 ijms-25-07750-t005:** Spinal cord organoids.

Author (Year)	Cell Source	Region Designed	Significance
Meinhardt et al. (2014) [67]	Mouse ESC	Neural tube	Spatial organization of neurons along the dorsal and ventral axis
Ranga et al. (2016) [68]	Mouse ESC	Neural tube	Focus on extracellular matrix and cytoskeleton during patterning
Lippmann et al. (2015) [70]	hiPSC	Neuromesoderm	Identified importance of HOX
Ogura et al. (2018) [71]	hiPSC	Dorsal and ventral spinal cord-like tissues	Dorsal and ventral tissues generated using BMP4 and Shh signaling
Andersen et al. (2020) [72]	hiPSC	Dorsal and ventral spinal cord-like tissues	A continuous dorsal and ventral tissue generated and connected muscle spheroids with neurons
Mouilleau et al. (2021) [73]	hESC/hiPSC	Rostro-caudal patterning	Differentiation of motor neurons through FGF and HOX expression
Karzbrun et al. (2021) [75]	hiPSC	Neural tube	Developed chip-based culture system for homogeneity
Lee et al. (2022) [27]	hiPSC	Spinal cord	Generated tubular structure with functionality
Shin et al. (2023) [76]	hPSC	Spinal cord	Necrotic core was reduced
Gribaudo et al. (2023) [28]	hiPSC	Spinal cord	Autonomously forming AP axis

## 3. CNS Organoids as Disease Models and Drug Screening Platforms

CNS organoids aid in understanding responses under normal or abnormal conditions, which is crucial for CNS disorder mechanisms, diagnosis, prevention, drug efficiency, and intraoperative imaging. They provide valuable insights into previously hard-to-study CNS diseases. This section discusses their use in disease modeling and advantages as CNS disease models.

### 3.1. Viral Infections

#### 3.1.1. Zika Virus (ZIKV)

Using organoids in viral infections brings benefits to researchers, as it can model the developmental process of CNS and mimic cellular diversity in vitro. The Zika virus (ZIKV), flavivirus, has been a significant threat since 1947, and re-emerged in 2015 [74]. Zika infection in the first trimester is a high-risk factor for newborn microcephaly. Garcez et al. [1] found that ZIKV impedes neural cell proliferation, reducing the number and size of neurospheres and brain organoids. A RT-qPCR analysis showed the upregulation of toll-like receptor 3 (TLR3) in ZIKV organoids, inducing apoptosis and disrupting neural progenitor cell development [77]. TLR3 inhibition in ZIKV organoids demonstrated its role in Zika-induced microcephaly, affecting neural progenitor cells (NPCs) that are heavily reliant on RNA interference for anti-viral immunity (Figure 3) [78].

#### 3.1.2. Herpes Simplex Virus (HSV)

Human-specific HSV infections require unique models, as HSV coevolves with human hosts. D’Aiuto et al. [79] used 2D and 3D organoids to replicate acute and latent HSV-1 infection, showing neuronal permissiveness and latency. Comparing HSV-1- and ZIKV-infected brain organoids revealed a distinct cellular architecture and effects, with HSV-1 causing clustered cell apoptosis and non-neural differentiation, while ZIKV disrupted the cell cycle and increased the stress and antiviral responses [80].

#### 3.1.3. Severe Acute Respiratory Syndrome-CoronaVirus-2 (SARS-CoV-2)

Organoid models quickly adapt to new viral studies, such as severe acute respiratory syndrome-coronavirus-2 (SARS-CoV-2) [81]. In August 2020, it was confirmed that SARS-CoV-2 can infect human neuronal cells using Cos [82]. SARS-CoV-2 targets neural progenitor cells and brain organoids, unlike SARS-CoV. This direct CNS infection may explain the prolonged anosmia, ageusia, and other reported neurologic symptoms. The Lancaster group found that SARS-CoV-2 infects choroid plexus cells, disrupting the blood–CSF barrier (Figure 3) [83]. CNS organoid models for viral infections are crucial for understanding infection mechanisms and accelerating solution development.

### 3.2. Neurodevelopmental Diseases

#### 3.2.1. Microcephaly

Microcephaly is an autosomal recessive neurodevelopment disorder resulting in significantly reduced brain size due to failures in neural stem cell maintenance, proliferation, migration, and cortical expansion [84]. Mutations in genes encoding proteins involved in centriole biogenesis, microtubule regulation, DNA damage response, signaling pathways, transcription, and metabolism contribute to the disorder [84,85]. Lancaster et al. [20] recreated an organoid lacking the cyclin-dependent kinase 5 regulatory subunit protein 2 (CDK5RAP2) protein, which lacked the neuroepithelial tissues containing progenitor cells, leading to increased neurons and decreased radial glia (RGs). The premature differentiation of progenitor cells into neurons prevented cortical expansion. Reintroducing the CDK5RAP2 protein restored neuroepithelial tissues, confirming its role in microcephaly. Li et al. [86] targeted the *abnormal spindle-like microcephaly-associated* (*Aspm*) gene, the most common mutation in human microcephaly. *Aspm*-mutant organoids failed to form structured neuroepithelial tissues and lacked ventricular radial glial (vRG) and outer radial glial (oRG) cells. These organoids had fewer *Pax6*, *SRY-related HMG-box genes 2* (*Sox2*), and tight junction proptein-1 (ZO-1) positive cells, which are crucial for neurodevelopment. Calcium imaging showed less neuron maturation and synchronized neuronal activities, correlating with congenital failure and mental retardation in patients with microcephaly. Zhang et al. [87] studied the WD repeat-containing protein 62 (WDR62) gene in COs. WDR62 deletion disrupted oRG proliferation and caused ciliogenesis defects, resulting in abnormally long cilia and more ciliated NPCs. Investigating the WDR62-centrosomal protein of the ~170kDa (CEP170)-kinesin family member 2A (KIF2A) pathway, KIF2A overexpression rescued the size defect in WDR62-deleted organoids. Asparaginyl-TRNA synthetase 1 (NARS1) loss also led to microcephaly [85]. The disease model showed reduced RG generation and poorly organized *Sox2^+^* cells. Reduced mRNA levels of the RG marker Pax6 and the mature neuronal marker microtubule-associated protein 2 (MAP2) were observed. Single-cell RNA sequencing (scRNAseq) revealed cell cycle defects and altered the cell fate towards astrocytes, decreasing organoid viability. Dhaliwal et al. [88] explored the role of phosphatase and tensin homolog (PTEN) in microcephaly. PTEN overexpression resulted in significantly smaller organoids with fewer Ki-67^+^ cells in the ventricular zone. Reduced NPC proliferation led to premature neuronal differentiation, indicated by high doublecortin (DCX) and transcription factor COUP TF1-interaction protein 2 (CTIP2) levels. In vitro, the PTEN-overexpressed organoid demonstrated reduced protein kinase B (AKT) activity, implicating the AKT-mTOR pathway in cerebral formation. Fair et al. [83] generated COs with an AUTS2T534P missense variant. These organoids showed reduced growth and pH3^+^ cells from day 20. Decreased CC3^+^ markers suggested compensatory mechanisms for reduced NPCs. Autism susceptibility candidate 2 (AUTS2)-COs confirmed the previous findings on cell cycle disruption in microcephaly, showing decreased NPC proliferation and increased asymmetrical divisions, leading to premature neuron generation and cell disorganization. While primary microcephaly involves numerous factors and multiple gene loci, the exact pathogenesis remains unclear.

#### 3.2.2. Autism Spectrum Disorder (ASD)

Autism spectrum disorder (ASD), a neuropsychiatric disorder, significantly affects human cognition and behavior. Current research models have limitations in classifying and studying the disease [89]. Urresti et al. [90] investigated the 16p11.2 copy number variation (CNV), the most common CNV associated with ASD. They selected 16p11.2 deletions (DEL) and duplications (DUP) in CNV carrier patients based on head circumference. DEL and DUP organoids matured into their respective phenotypes, showing significant differences from the controls. A gene ontology (GO) analysis of transcriptomes and proteomes at 1 and 3 months revealed dysregulated genes related to the actin cytoskeleton, neuron function, ion channels, synaptic signaling, and nervous system development. Impaired cortical neuron migration in 16p11.2 organoids was associated with Ras homolog family member A (RhoA) signaling. de Jong et al. [91] modeled the forebrain ASD organoid with the c.3709DelG mutation in contactin associated protein 2 (CNTNAP2), showing abnormal cell proliferation and differentiation, confirming early embryonic development impact, and leading to early brain growth and macrocephaly. Jourdon et al. [92] compared macrocephalic and normocephalic ASD patient-derived COs. Both groups had similar differentiation rates, but macrocephalic ASD organoids upregulated self-renewal genes and downregulated differentiation genes. Macrocephalic ASD organoids showed an increased production of cortical plate excitatory neurons, while early preplate and inhibitory neuron production decreased. Normocephalic ASD organoids demonstrated increased early preplate neuron production and decreased cortical plate excitatory neurons. This indicates that macrocephalic ASD progenitors favor RG expansion and cortical plate excitatory neuron neurogenesis. Li et al. [86] developed the CRISPR–human organoids–single-cell RNA sequencing (CHOOSE) system, using guide RNAs and CRISPR-Cas9 for pooled loss-of-function screening in mosaic organoids. This model enabled the study of ASD gene loss of function and developed a phenotypic database for ASD. It revealed that certain cell types, such as dorsal progenitors and L2/3 excitatory neurons, are more susceptible to ASD perturbations than others.

#### 3.2.3. Developmental and Epileptic Encephalopathy (DEE)

Developmental and epileptic encephalopathy (DEE) impairs cognitive functions due to seizure and interictal epileptiform activity [93]. Hengel et al. [94] found that UDP-glucose 6-dehydrogenase (UGDH) germline mutations, which act as loss-of-function alleles, caused significant underdevelopment in organoids. The knock-out of UGDH hindered FGF signaling, identifying UDP-GlcA as a potential therapeutic target. Borghi et al. [95] studied protocadherin 19 (PCDH19)-related epilepsy, revealing increased neuronal differentiation and neurite length due to PCDH19’s role in mitotic spindle formation. In COs, WT cell proliferation decreased, while neurogenesis increased, leading to smaller organoids.

WWOX-related epileptic encephalopathy (WOREE), a severe form of DEE, was modeled using WWOX CRISPR-edited and patient-derived COs, showing electrophysiologically, hyperexcitability [96]. The pathophysiology revealed increased GABAergic and astrocytic markers, indicating a disrupted neuronal network balance. Impaired DDR signaling in WWOX-depleted organoids displayed increased astrocytic markers, possibly from vRGs. Autosomal recessive spinocerebellar ataxia-12 (SCAR12) phenotype organoids showed similar results, indicating cortical origins of the WWOX deficiency.

Negraes et al. generated cyclin-dependent kinase-like 5 (CDKL5) deficiency disorder (CDD) COs from six patients to study its pathogenesis [97]. The analysis revealed CDKL5’s role in defective NPC proliferation, mTOR pathway hyperactivation, and altered neuronal connectivity and excitability. CDD NPCs showed slower proliferation, increased cell death, neurons with longer dendrites, increased spine densities, and impaired synaptogenesis. Proteomics indicated mTOR hyperactivation and elevated α-amino-3-hydroxy-5-methyl-4-isoxazolepropionic acid receptor (AMPAR) and mGluR levels, with cells more susceptible to hyperexcitability.

Lee et al. [27] tested anti-epileptic drugs (AEDs) on morphogenesis using an automated analysis system. SCOs were exposed to different concentrations of six AEDs, including valproic acid, carbamazepine, phenytoin, primidone, lamotrigine, and gabapentin, to observe neural tube defects (NTDs). Valproic acid- and carbamazepine-treated hCOs displayed NTDs due to delayed or failed tube closure, with significant morphological defects and delayed neural tube development and zippering.

### 3.3. Neurodegenerative Diseases

Neurodegenerative diseases cause the progressive loss of specific neuronal subsets, leading to motor, sensory, and cognitive defects, and are becoming more common as lifespans increase [98].

#### 3.3.1. Alzheimer’s Disease (AD)

Alzheimer’s disease (AD) is characterized by memory impairments, brain atrophy, neuronal loss, amyloid plaques, and neurofibrillary tangles (NFTs). Raja et al. [99] created neural organoids from patients with familial AD, showing amyloid pathology before tau hyperphosphorylation. The inhibition of APP processing reduced both amyloid-beta (αβ) accumulation and tauopathy. Gonzalez et al. [100] used COs to demonstrate that cellular apoptosis in AD is proportional to protein aggregates, which resemble senile amyloid plaques and NFTs. The aggregation of αβ and tau proteins is proportional in AD, with αβ aggregates corresponding to the greatest tau abnormalities. Ghatak et al. [101] explored the link between AD and neuronal hyperexcitability, finding that AD models exhibited significant hyperexcitability and reduced neurite length. Alic et al. [102] investigated the role of β-secretase (BACE) in patients with Down syndrome (DS) with AD, revealing that the β-secretase activity of BACE1 correlates directly with αβ levels, whereas BACE2 activity does not. Single nucleotide polymorphisms (SNPs) at the BACE2 locus influence the age of AD onset in the DS population. The trisomy 21 (T21) organoids, conditioned media (CM), and DS cerebrospinal fluid (DS-CSF) showed an increased ratio of BACE2-αβ degradation products to amyloidogenic and α-site products. BACE2 degrades αβ, protecting against AD. A reduction in the BACE2 copy number led to early AD-like pathology without amyloid plaques in T21 COs. Eliminating the third BACE2 copy in T21 resulted in AmyloGlo^+^ deposits with neuritic plaques and neuron loss, while the trisomic BACE2 level protected against early amyloid plaque pathology, suggesting BACE2 as a therapeutic target to delay AD onset. Zhao et al. [103] examined the phenotypes and degenerative pathways in AD caused by APOE4 allele, a genetic risk factor for late-onset AD. AD organoids exhibited elevated αβ-40 and αβ-42 levels in RIPA-soluble fractions. The isogenic conversion of APOE4 to APOE3 reduced αβ levels. Although ELISA and immunostaining did not detect insoluble αβ and amyloid plaques at week 12, all AD organoids showed increased p-tau levels. APOE4 accelerated p-tau accumulation independently of αβ, as neither APOE4 nor apoE levels influenced αβ levels. Mature AD organoids exhibited reduced synaptic proteins and increased apoptosis. The AD condition itself had the most significant impact, while APOE4 did not strongly affect the profile. Chen et al. [104] aimed to replicate the blood–brain barrier (BBB) breakdown associated with age-related AD using human serum in COs. The organoids recapitulated the key AD pathology characteristics. Serum exposure increased αβ and p-tau levels by elevating BACE and glycogen synthase kinase-3 (GSK3) α/β expression, disrupting synaptic gene expression in neurons and astrocytes. Inhibitors targeting αβ production or p-tau reduction did not prevent synaptic loss. Park et al. [3] developed a model integrating mathematical modeling and COs to facilitate precision medicine. This model presented a molecular regulatory network that accounted for neuronal dynamics. The sex-dependent characteristics of AD pathogenesis can be studied using organoid models, which is complicated under clinical investigations [105]. The researchers generated CO from healthy patients and modeled AD using the αβ peptide. Then, estrogen was treated to the AD organoid model, in which αβ peptide was suppressed upon the treatment of estrogen. Flannagan et al. [106] rather directly compared the effect of sex on AD, specifically focusing on the phenotype of mitochondria using COs from different sexes. It was found that mitochondrial dysfunctions are largely sex-dependent, leading to problems in proteostatsis.

#### 3.3.2. Parkinson’s Disease (PD)

Parkinson’s disease (PD) is characterized by motor symptoms, such as parkinsonism, which is associated with Lewy bodies and the loss of dopaminergic neurons in the substantia nigra, as well as non-motor features involving the nervous system, neurotransmitters, and protein aggregates [107]. The exact cause of PD remains unknown, but it appears to result from a combination of genetic and environmental factors affecting numerous cellular processes [108]. Kim et al. [2] generated leucine-rich repeat kinase 2 (LRRK2)-G2019S midbrain organoids to model LRRK2-based late-onset PD. They found that the upregulation of thioredoxin-interacting protein (TXNIP) and its suppression inhibited LRRK2-induced phenotypes. Smits et al. [109] produced a CO model displaying PD phenotypes and midbrain dopaminergic neurons (mDANs) by expanding midbrain floor plate neural progenitor cells (mfNPCs). The LRRK2-G2019S mutation led to a reduced number and complexity of mDANs, with a significant increase in forkhead box protein A2 (FOXA2) progenitor cells, suggesting a compensatory response to impaired mDAN specification. Reversing the LRRK2-G2019S mutation did not salvage the organoid from exhibiting the PD phenotype. Kwak et al. [54] developed a midbrain organoid that captured the essential mDAN and other functional cell types. Using dorsomorphin, A83-01, and CHIR99021 (DAC 3.0), they produced midbrain organoids with midbrain features, including laminated structures, the homogeneous distribution of mature mDAN, and the production of neuromelanin-like granules. DAC 3.0-treated midbrain organoids exhibited cell death upon treatment with the neurotoxin 1-methyl-4-phenyl-1,2,3,6-tetrahydropyridine (MPTP), indicating that they can recapitulate the PD pathophysiology mediated by astrocytes and mDANs. Wulansari et al. [110] generated a midbrain organoid with DnaJ heat shock protein family (Hsp40) member C6 (DNAJC6) function mutations, which impaired ventral midbrain (VM) patterning and caused defects related to adult mDANs, leading to neurodegeneration and other PD phenotypes. The analysis revealed that the loss of DNAJC6 led to PD pathogenesis by affecting the transcription of mDAN-specific genes and disrupting the vesicular trafficking of lysosomal proteins. Bercerra-Calixto et al. [111] produced midbrain organoids displaying increased pathological alpha-synuclein (αSyn), Lewy body (LB)-like inclusions, and neurodegenerative alterations similar to patients with PD. Although pS129^+^ αSyn was immunoreactive in organoids, biochemical assays could not consistently detect pS129^+^ αSyn. Comparing PD–midbrain organoids with brain sections in a Lewy body dementia (LBD) patient revealed similar Lewy bodies and αSyn inclusions in morphology, intracellular localization, and marker composition. PD-like degeneration of mDANs was evident through tyrosine hydroxylase and cleaved caspase-3. The limitations included the immaturity of PD organoids, as αSyn inclusions occurred more frequently with maturation, and the absence of core and pS129-positive lamellar bands. The loss-of-function mutation in PTEN-induced kinase 1 (PINK1) caused PD by affecting the transition between the NPCs and TH^+^ neurons.

#### 3.3.3. Huntington’s Disease (HD)

Similarly, Huntington’s disease (HD) is a progressive genetic disorder that causes the breakdown of nerve cells in the brain and, generally, its progression is known to be faster than PD. However, compared with PD, the studies on HD using organoids are not as rigorous because the number of patients is small. Direct genetical analyses (e.g., CAG repeat [112] and CHCHD2 [113]) on HD pathogenesis using organoids are reported by a few research groups. Metzger et al., on the other hand, employed the deep learning technique to identify the phenotypes of micropatterned organoids and applied these to screen drugs for HD [114]. The study demonstrated that the organoid-based drug-screening platform is effective in discovering new drugs for diseases.

#### 3.3.4. Motor Neuron Diseases

Motor neuron diseases are a subset of neurodegenerative disease affecting motor neurons, spinal sensory neurons, and the muscular system [115,116]. Pereira et al. [117] generated an amyotrophic lateral sclerosis (ALS) patient-derived sensorimotor organoid to elucidate the ALS mechanism involving neuromuscular junctions (NMJs). Six isogenic lines displayed a reduced variation in sphere composition and derived cell types. Comparing the pairs of isogenic lines with the familial ALS mutations revealed NMJ defects. Specific ALS-related phenotypes were observed: *SOD1G85R* and *PFN1G118V* mutations showed a reduced percentage of innervated NMJs, while *TARDBPG298S* mutations exhibited a decreased area of innervated NMJs. Tamaki et al. [118] examined the effects of the TAR DNA-binding protein 43kDa (TDP-43). Patient-derived protein extracts with pathogenic TDP-43 replicated ALS pathology in COs, causing astrogliosis, cellular apoptosis, and genomic damage. COs from patients with sporadic ALS exhibited increased TDP-43 aggregates and pathology six weeks earlier than controls, suggesting that unknown factors in patients with ALS make them more susceptible to TDP-43 pathology. The upregulation of glial fibrillary acidic protein (GFAP) confirmed astrogliosis. ALS patient-derived protein extracts induced cell apoptosis and double-stranded breaks in recipient COs, linking cellular death to genomic damage in TDP-43 pathology. Negi et al. [119] produced a TDP-43 organoid model of ALS. A proteomic analysis identified 126 deregulated proteins, with 44 upregulated and 82 downregulated, significantly impacting normal cellular functions and providing numerous potential therapeutic targets against ALS. Chow et al. [120] developed a motor nerve organoid to model axonal degeneration using custom-made tissue culture devices. These motor organoids featured cell bodies in a spheroid with a unidirectional axonal bundle and were treated with hydrogen peroxide to emulate degeneration.

Despite the intriguing results from CNS organoid studies on neurodegenerative diseases, questions remain about their effectiveness, particularly in the long-term studies required to accurately reproduce the natural aging process.

### 3.4. CNS Tumors

Central nervous system (CNS) tumors can be broadly classified into primary and secondary types. Primary CNS tumors originate from cells within the CNS, while secondary tumors metastasize from other parts of the body. Despite advancements in medical and surgical treatments, the recurrence rate remains high, leading to significant mortality [121,122].

#### Glioblastoma

Glioblastoma is the most common and lethal CNS tumor in adults [123]. Ogawa et al. [124] created COs patient-derived glioblastoma by co-injecting plasmids containing pSpCas9(BB)-2A-GFP and a TP53 single-guide RNA (sgRNA). The patient-derived tumor was generated using a donor plasmid with an oncogenic cytomegalovirus (CMV) promoter-driven HRasG12V, disrupting the *TP53* gene. IRES-TdTomato was introduced as a fluorescent marker and to monitor tumor cells within the oncogenic cassette. Two weeks post-expression, both the CRISPR/Cas9 and TdTomato-HRASG12V organoids exhibited GFP expression (Figure 4). The control group showed minimal Ki-67 activity at 11 weeks, whereas the disease models had increased Ki-67^+^ cells and a higher expression of the brain tumor stem cell markers, SOX2 and OLIG2. The molecular profile indicated a mesenchymal subtype of glioblastoma. In vivo testing involved the stereotaxic injection of organoid-derived tumor cells into NOD/SCID/IL2RG-KO mice, resulting in death within months, with H&E staining showing glioblastoma characteristics. Tumors were transplanted between organoids retaining tumorigenic markers. Linkous et al. [125] generated glioblastoma COs by co-culturing CO with GFP-labeled patient-derived glioma stem cells, achieving a 100% infiltration rate. These COs retained parental tumor characteristics and replicated microtubule formation, supporting tumor growth and necrosis. da Silva et al. [126] developed an ex vivo glioblastoma by culturing mouse COs with glioblastoma spheroids, comparing it with the COs co-cultured with NPCs. A timelapse microscopy showed the incorporation into mouse embryonic stem cell COs. Krieger et al. [127] quantitatively and transcriptionally analyzed glioblastoma using an in vitro model, co-culturing COs with fluorescently labeled glioblastoma cells from four patient-derived lines. Glioblastoma cells invaded the neuronal layers but not the neural progenitor rosettes and did not invade organoids derived from the breast cancer cell line MCF10AT or the neuroblastoma cell line SH-SY5Y. Imaging confirmed heterogeneity in the invasion depth and microtubule length, promoting tumor dissemination. A principal component analysis (PCA) and t-distributed stochastic neighbor embedding (t-SNE) maps confirmed intertumoral heterogeneity and differential subtype expression across patient samples. A differential gene expression (DEG) analysis showed that CO-–glioblastoma interaction enhanced gene expression for tumor network formation and invasion. Identified ligand–receptor pairs included glutathione S-transferase P/tumor necrosis factor receptor-associated factor 2, Notch receptors 1 and 2/DLK1, and collagen/integrins, suggesting their role in glioblastoma progression. Sundar et al. [128] compared the advantages and disadvantages of spheroids and COs in studying the clinical application of glioblastoma. Organoids generated from adult and pediatric surgical specimens showed consistent proliferation in the organoid rim, a feature not seen in spheroids or adherent cultures.

Medulloblastoma is the most aggressive and lethal pediatric CNS tumor, with a limited understanding of its biomechanism and effective treatments [129]. Ballabio et al. [130] generated cerebellar organoids to elucidate the mechanisms of medulloblastoma. Among the medulloblastoma subtypes, group 3 is the most aggressive. In vivo studies revealed that the overexpression of the *Gfi1 + c-MYC (GM)* and *Otx2 + c-MYC (OM)* genes produced group 3 medulloblastoma. Signaling pathways were examined as potential drug targets, identifying SMARCA4 as the most frequently mutated site. The overexpression of SMARCA4 reduced medulloblastoma expression. Electroporation with GM and OM resulted in small buds of Venus^+^ and PCNA^+^ cells, indicating the overproliferation and impaired differentiation of cerebellar progenitors (Figure 4). SMARCA4 overexpression reduced OM-induced medulloblastoma but was less effective against GM-induced medulloblastoma. The medulloblastoma model induced medulloblastoma in vivo after injection, and the global DNA methylation profiles of OM- and GM-injected organoids closely resembled patient-derived medulloblastoma. van Essen et al. [131] explored the PTCH1-mutant pathway to generate a cerebellar medulloblastoma model. A differentiation observation showed significantly higher EN1 genotype expression in PTCH1+/- and PTCH1-/- compared with the controls. PTCH1-/- organoids expressed significantly lower levels of Pax6, ATOH1, and KIRREL2, with increased Shh pathway activity through the upregulation of GLI1. Homozygous organoids had upregulated ventral neural tube genes and downregulated dorsal neural tube genes. These results indicated that the loss of PTCH1 function led to ventralization and a shift from mid-hindbrain to forebrain formation.

While CNS organoid studies offer promising insights, challenges remain, especially for the long-term studies necessary to replicate natural aging and tumor progression. These models, however, provide valuable pathways for therapeutic development and personalized medicine.

## 4. Future Perspectives and Conclusions

Over the past decade, significant advancements have been made in the development of CNS organoids. From the early, coarse COs, researchers can now generate region-specific CNS organoids by studying the advancements in morphogenetic processes. These developments have clarified fabrication methods and expanded their applications in CNS disease modeling, promising substantial innovation in treating various CNS diseases.

Despite these advancements, CNS organoids face clear limitations. Unlike the organoids of other organs, current CNS organoids do not fully satisfy the definition of “organoid”, which includes features such as a relevant physiological microenvironment, multiple cell types, and self-assembly [132].

Current CNS organoids fail to demonstrate these characteristics comprehensively. While COs display various types of neural cells that self-layer in the cortex, they lack the specific sub-regional organization and structural integrity necessary for interregional interactions within the CNS. Although researchers have succeeded in creating region-specific and vascularized brain organoids, they still lack the structural integrity of a whole brain. Moreover, using CNS organoids as drug-screening platform requires detailed study before clinical application. Current organoids can only capture cell-scale reactions, determining the dose-dependent cellular toxicity or effectiveness. In both the pre-clinical and clinical settings, considering the interactions of the CNS with other organs (e.g., the liver and kidney) is essential for assessing drug efficacy and dosage. Therefore, the standards for drug testing with CNS organoids need to be established for the further application of them in the pharmaceutical industry.

Large-scale fabrication methods must accompany the advancement in CNS organoids. Lab-scale fabrication has limitations for clinical use. Among these limitations, ethical issues present a significant challenge that is unique to CNS organoids [133]. Most current brain organoids are grown without external sensory stimuli, differing from in vivo brain development. This raises ethical questions about the potential of brain organoids possessing consciousness. The controversies are difficult to define and settle, necessitating a social and scientific consensus to advance the field of CNS organoids.

Several intriguing topics remain unexplored in CNS organoid research. For example, the connection between CNS organoids and other organs should be studied. By examining these connections, researchers can gain insights into physiological mechanisms and related diseases. An organ-on-a-chip approach could be ideal for studying the interactions between CNS organoids and other organoids, broadening our understanding of organ–organ interactions (Figure 5) [134]. Additionally, the microenvironments of CNS organoids require further study. While Matrigel is commonly used, the physical properties of growth media can alter stem cell characteristics, suggesting that other media might be more suitable. The effects of extrinsic growth factors on CNS organoid growth are also not as well understood, as are their effects on other organoids. Given the complexity of CNS development, investigating various growth factors’ impacts on CNS organoid growth will be crucial for understanding neurodevelopment.

Organoids hold great promise for studying human CNS development in vitro and provide a platform for investigating diseases previously difficult to study. As their applications in drug screening and regenerative medicine are invaluable, the demand for CNS organoids is growing. Although hurdles remain, CNS organoids will guide researchers into uncharted territories of human CNS research.

## Figures and Tables

**Figure 1 ijms-25-07750-f001:**
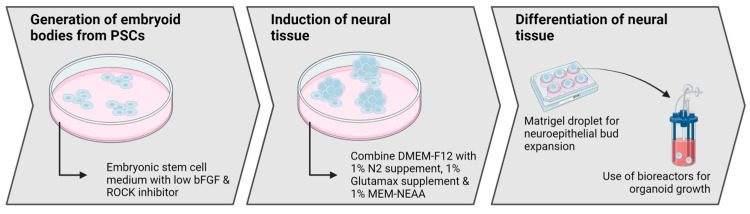
The landmark study by Lancaster et al. set the standard for developing cerebral organoid (CO) protocols. Embryoid bodies are generated successfully from pluripotent stem cells (PSCs) using decreased basic fibroblast growth factor (bFGF) and a high-dose rho-associated protein kinase (ROCK) inhibitor. Then, the neural tissue is inducted by altering the media. Matrigel droplets and a bioreactor are used to expand and grow COs. Subsequent studies sought to enhance and lessen the limitations of the protocol.

**Figure 2 ijms-25-07750-f002:**
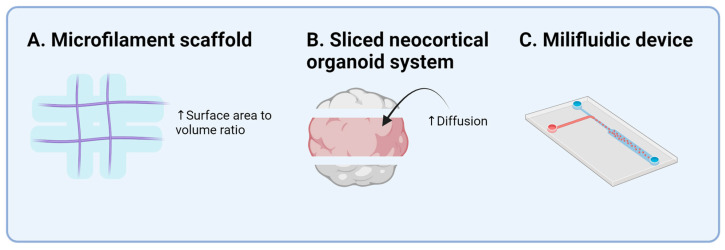
Schematic figure describing the technology to overcome the limitations mentioned in Figure 1. (**A**) Microfilament scaffold provides embryoid bodies to attach and enlarge the surface-area-to-volume ratio. (**B**) Sliced neocortical organoid system slices an organoid to allow greater diffusion. (**C**) Milifluidic device allows for efficient exchange of culture media and waste. The application of this technology enables the organoid to expand homogeneously with a smaller necrotic core.

**Figure 3 ijms-25-07750-f003:**
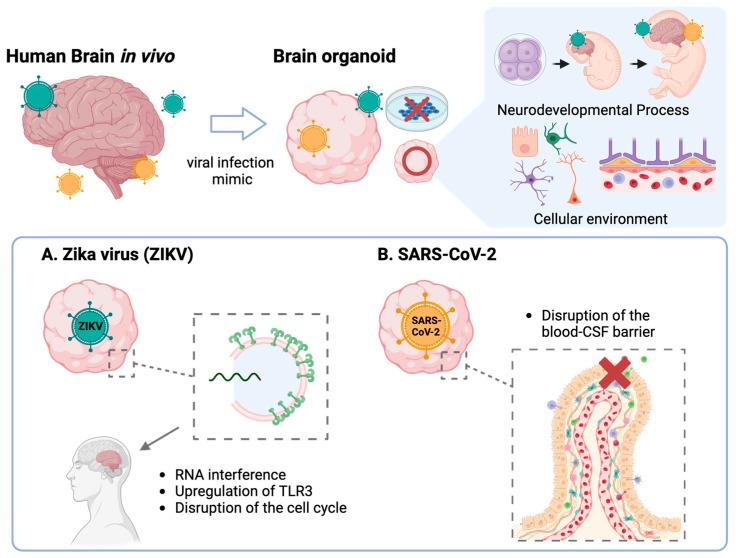
Schematic illustration of the advantages of using organoids in CNS viral infection, compared with the conventional 2D viral infection model. (**A**) Zika virus infection causes RNA interference, upregulation of toll-like receptor 3 (TLR3) and disruption of the cell cycle in neural progenitor cells leading to microcephaly. (**B**) Severe acute respiratory syndrome-coronavirus-2 (SARS-CoV-2) infects choroid plexus cells, affecting the blood–cerebrospinal fluid barrier and causing neurological symptoms.

**Figure 4 ijms-25-07750-f004:**
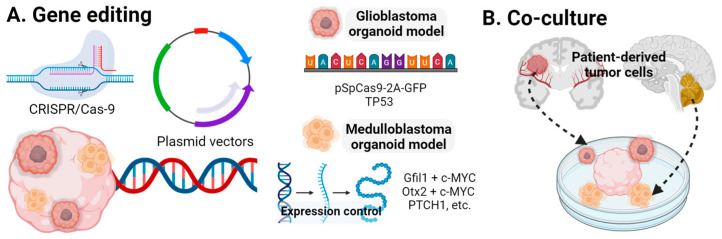
Schematic illustration of the generation of central nervous system (CNS) tumors using organoids. (**A**). The glioblastoma organoid was generated using a combination of CRISPR/Cas-9 and plasmid methods as pSpCas9-2A-GFP and TP53 single-guide RNA was inserted within the plasmids. Medulloblastoma using CRISPR/Cas-9 explored several genes, such as Gfil1 + c-MYC, Otx2 + c-MYC, and PTCH1. (**B**). Co-culturing with the patient-acquired surgical specimen also created disease models.

**Figure 5 ijms-25-07750-f005:**
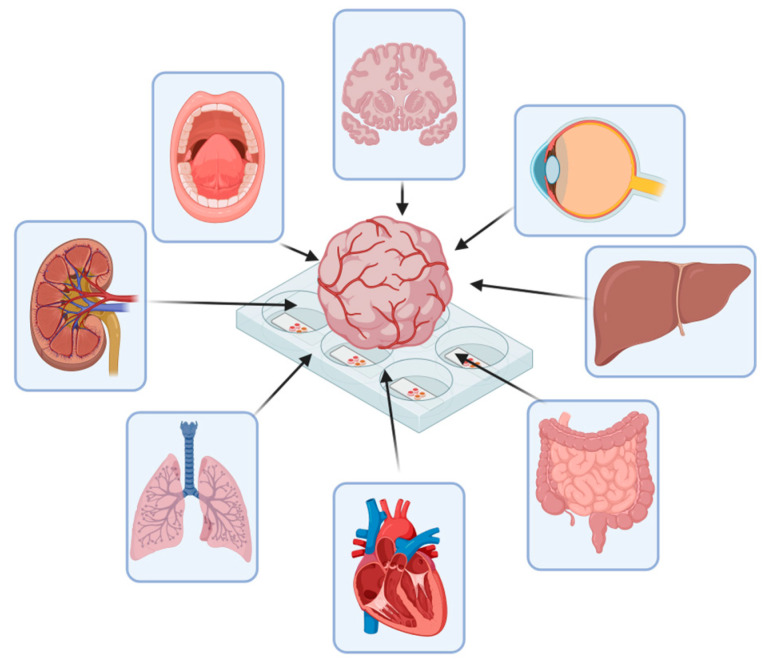
Schematic illustration of possible organ-on-a-chip approach in the future.

**Table 1 ijms-25-07750-t001:** Cortical organoids.

Author (Year)	Cell Source	Region Designed	Significance
Lancaster et al. (2013, 2014) [20,31]	hPSC	Heterogenous	-First organoid -Designed microcephaly
LaMonica et al. (2013) [33]	Fetal brain tissue	Neocortex, radial glia	Discovery of radial glia division and the involved pathways
Quadrato et al. (2017) [34]	hPSC	Cerebral cortex	Prolonged culture of cerebral organoids led to presented diverse cells
Lancaster et al. (2017) [35]	hiPSC/hESC	Cerebral cortex	Reproducible cerebral organoids with microfilament device
Velasco et al. (2019) [36]	hiPSC	Cerebral cortex	Developed consistent ways to generate organoids
Pollen et al. (2019) [37]	hiPSC/Chimpanzee-induced PSC	Cerebral cortex	Investigation of organoids among species to analyze differences
Qian et al. (2020) [38]	hiPSC	Cerebral cortex	Developed sliced neocortical organoid system
Uzquiano et al. (2022) [39]	hESC/hiPSC	Cerebral cortex	Single cell analysis of multiple organoids derived from multiple cell sources

**Table 2 ijms-25-07750-t002:** Forebrain organoids.

Author (Year)	Cell Source	Region Designed	Significance
Krefft et al. (2018) [41]	hiPSC	Early dorsal telencephalic tissues	First forebrain organoid
Bagley et al. (2017) [42]	hiPSC	Fusion of dorsal and ventral forebrain organoid	-Production of dorsal and ventral forebrain organoid -Ability to study neurite dynamics
Cederquist et al. (2019) [43]	hPSC	Anterior forebrain	Positional identity defined
Gabriel et al. (2023) [44]	hiPSC	Forebrain with eye primordium	Study of multi-organ association
Huang et al. (2021) [45]	hiPSC	“Arcuate organoids”	First efforts to create diencephalon-type forebrain organoid
Blaess et al. (2014) [46]	hiPSC	Hypothalamus	Hypothalamus development
Corman et al. (2018) [47]	Mice ESC	Hypothalamus	Elucidated Shh signaling and its effects with time
Sarrafha et al. (2023) [48]	hPSC	Hypothalamus	Exploration of dopaminergic neurons
Xiang et al. (2019) [49]	hESC	Thalamus	Fusion of cortex with thalamus organoid
Kiral et al. (2023) [50]	hESC	Thalamus and Thalamic reticular nucleus	Generation of thalamic reticular nucleus

**Table 3 ijms-25-07750-t003:** Midbrain organoids.

Author (Year)	Cell Source	Region Designed	Significance
Jo et al. (2016) [52]	hPSC	Midbrain	Developed midbrain organoids with dopamine neurons that form synapses and display electrophysiological properties
Monzel et al. (2009) [53]	Neuroepithelial stem cells	Midbrain	Glial cell differentiation
Kwak et al. (2020) [54]	hESC	Midbrain	Defining protocols to produce homogenous midbrain organoids
Nickels et al. (2020) [55]	hiPSC	Midbrain	Further tested protocols to generate organoids and reduce necrotic cores
Berger et al. (2018) [56]	Neuroepithelial stem cells	Midbrain	Introduced milifluidic device to reduce necrotic regions in long term culture
Renner et al. (2020) [57]	hPSC	Midbrain	Generated automated system for seeding, imaging, and analysis

**Table 4 ijms-25-07750-t004:** Cerebellar organoids.

Author (Year)	Cell Source	Region Designed	Significance
Kamei et al. (2023) [59]	hiPSC	Cerebellum	Investigated cerebellar progenitors through cell transplantation
Muguruma et al. (2015, 2018. 2020) [58,60,61]	hiPSC	Cerebellum, Purkinje cells	Ability to model cerebellar diseases
Silva et al. (2020) [63]	hiPSC	Cerebellum	Use of a single bioreactor to mass-produce homogenous cerebellar organoids
Nayler et al. (2021) [64]	hiPSC	Cerebellum	Air–liquid interface culture without SDF1 and FGF19
Chen et al. (2023) [8]	hiPSC	Cerebellum	Long-term culture of cerebellar organoids
Atamian et al. (2024) [65]	hiPSC	Cerebellum	Analysis of cellular composition and spatial characteristics

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
