# Peer review of "Morphogenetic Designs, and Disease Models in Central Nervous System Organoids"

_ijms, 2024, doi:10.3390/ijms25147750_

Round 1

Reviewer 1 Report

Comments and Suggestions for Authors

This review by Minsung Bok and colleagues about CNS organoids is very informative, detailed and well organized. I have very few comments/questions.

I know the review is already extensive, but it might be interesting to briefly touch on Huntington's disease (for example see https://doi.org/10.1016/j.crmeth.2022.100297).

As we know, sex can be a risk factor in some diseases, for example Alzheimer's disease. Is it possible to study the influence of sex using organoids?

-minor comments:

A word seems to be missing line 92: perhaps "CNS organoids help us understand CNS structure development...".

I noted that lines 98 to 106 were a repeat from the previous paragraph.

Line 285: please correct "reducing the number size of neurospheres...". Is it "number and size"?

Line 555: "suggesting their role IN glioblastoma progression." 

Comments on the Quality of English Language

Very minor English editing is needed in my opinion.

Author Response

Thank you so much for your constructive review, we've attached the response as a word file.

Reviewer 2 Report

Comments and Suggestions for Authors

This review provides a valuable resource for researchers in the field of CNS organoids. With some minor revisions and enhancements, it has the potential to be a highly informative and influential paper in the field. The authors are encouraged to incorporate the suggested changes to improve the clarity, depth, and overall impact of their work.

Major Comments:

1.     The paper provides a comprehensive overview of CNS organoids, covering a wide range of organoid types and their applications. However, the inclusion of more detailed comparisons between different protocols and their efficiency could enhance the depth of the review.

2.     While the paper is well-organized, some sections could benefit from clearer subheadings and transitions. For example, the section on disease models could be subdivided into specific diseases for better readability.

3.     The methodologies described are thorough but could be supplemented with more figures or diagrams to illustrate key protocols and findings. This would aid in better understanding complex processes.

4.     The figures are not well presenting the advances in organoid study and did not coordinate or help to illustrate the text description.

Minor Comments

Some statements would benefit from additional references to support the claims made. For instance, the discussion on the reproducibility of organoids would be strengthened by citing specific studies.

Comments on the Quality of English Language

Ensure all typographical errors are corrected. For instance, there are occasional issues with punctuation and grammar that should be addressed for a smoother read.

Author Response

Thank you so much for taking the time to give us a through review. We have adapted all of the suggestions and have attached a file below with the changes.
